# Generalized Dual-Scale Optimization: Topology-Aware Margin Dynamics in Fine-Grained Vision

**Lingfeng Xia**
Independent
xlfkaixin@gmail.com

## Abstract

We identify that the standard Cross-Entropy loss exhibits a monotonically expanding intrinsic margin, causing gradient saturation in fine-grained tasks. To address this, we propose the Generalized Dual-Scale Loss, a unified framework controlling margin dynamics via a parameter $\lambda$. Experiments with Vision Transformers reveal that optimal dynamics are topology-dependent: rigid, geometric manifolds require aggressive hard mining ($\lambda > 1$) to resolve structural subtleties, whereas noisy, biological manifolds favor robust constant margins ($\lambda \approx 1$) to prevent overfitting to clutter. Our work advocates for aligning optimization dynamics with the intrinsic noise and granularity of the data.

## 1 Introduction

Cross-Entropy (CE) is the canonical objective for deep classification; beyond maximum likelihood, it can exhibit an implicit bias toward max-margin solutions in separable settings (Soudry et al., 2018). However, for fine-grained recognition, we argue that CE can "brake" too early by introducing a confidence-dependent smoothing buffer.

Geometrically, the ideal constraint requires the target logit $x_y$ to exceed the strongest competitor $M = \max_{j \neq y} x_j$, which is captured by the hinge loss

$$\mathcal{L}_{\text{hard}} = \max(0, M - x_y). \tag{1}$$

CE can be viewed as a smooth relaxation of this objective,

$$\mathcal{L}_{\text{CE}} = \log\left(\sum_k e^{x_k}\right) - x_y. \tag{2}$$

We define the intrinsic margin (smoothing artifact) as

$$\Delta(x_y) = \mathcal{L}_{\text{CE}} - (M - x_y). \tag{3}$$

As confidence increases, this buffer can dominate optimization, reducing effective gradient pressure on subtle inter-class conflicts.

To control this behavior, we propose the Generalized Dual-Scale Loss (GDS), which decouples margin dynamics from confidence via a continuous parameter $\lambda$, interpolating between robust constant-margin behavior ($\lambda \approx 1$) and aggressive hard-mining dynamics ($\lambda > 1$).

**Positioning w.r.t.**
**prior work.** GDS complements approaches that re-weight gradients by confidence (Focal Loss (Lin et al., 2017)), enforce explicit geometric margins (ArcFace (Deng et al., 2019)), soften targets for calibration (label smoothing (Szegedy et al., 2016)), or focus on hard negatives (OHEM (Shrivastava et al., 2016)), by providing a single continuous control of margin dynamics matched to data topology.

## 2 METHODOLOGY

In this section, we formalize the Generalized Dual-Scale Loss. Our goal is to construct a unified objective that allows explicit control over the gradient dynamics relative to prediction confidence, enabling alignment with the underlying data topology.

### 2.1 THE GENERALIZED OBJECTIVE

Let $\mathbf{x} \in \mathbb{R}^K$ denote the logit vector and $y$ be the target class index. We first define two potential energy terms.

**Global Energy ($S_{\mathbf{all}}$).** The standard log-sum-exp over all classes: $S_{\text{all}} = \log \sum_k e^{x_k}$.

**Hard-Target Energy ($S(M)$).** To construct a robust geometric anchor, we define $M = \max_{j \neq y} x_j$ as the maximum non-target logit. Crucially, we treat $M$ as a detached constant (i.e., stop-gradient) with respect to the target $x_y$. The hard-target energy is defined as $S(M) = \log \left( \sum_{j \neq y} e^{x_j} + e^M \right)$.

**Note.** By explicitly adding $e^M$ to the sum of negatives, this term effectively up-weights the hardest negative (doubling its contribution in the limit), providing an implicit hard-mining emphasis in the spirit of OHEM (Shrivastava et al., 2016) even before $\lambda$ is applied.

Based on these energies, we define two component losses.

**1. The Soft Constraint ($L_{\text{right}}$).** This term corresponds to the standard Cross-Entropy loss:

$$L_{\text{right}} = S_{\text{all}} - x_y. \tag{4}$$

**2. The Adaptive Hard Constraint ($L_{\text{left}}$).** This term introduces the control parameter $\lambda$. It penalizes the distance to the hard-target energy, with an adjustable dependency on the global energy:

$$L_{\text{left}} = \lambda S(M) + (1 - \lambda)S_{\text{all}} - x_y. \tag{5}$$

**3. Temperature-Coupled Fusion.** To avoid the optimization instability associated with sparse gradients (typical of pure hinge loss), we fuse these components via a smooth soft-maximum gate (with $\tau = 1$ throughout):

$$\mathcal{L}_{\text{GDS}} = \log \left( e^{L_{\text{left}}} + e^{L_{\text{right}}} \right). \tag{6}$$

When the model makes large errors ($L_{\text{left}} \gg L_{\text{right}}$), the loss (and gradient) is dominated by the hard constraint; as it converges, it smoothly relaxes toward the soft constraint.

### 2.2 THEORETICAL VALIDATION: CONTROLLING MARGIN MONOTONICITY

To verify our design, we analyze the intrinsic margin behavior. As noted in the Introduction, standard CE exhibits a property where its margin expands monotonically with $x_y$ under fixed-competitor assumptions.

We define a $\lambda$-controlled intrinsic margin $\Delta(x_y; \lambda)$ as the deviation of our hard constraint from the geometric gap $(M - x_y)$:

$$\Delta(x_y; \lambda) = L_{\text{left}} - (M - x_y). \tag{7}$$

Substituting the definition of $L_{\text{left}}$ and noting that $\frac{\partial S(M)}{\partial x_y} = 0$ (since $M$ is treated as a detached constant, i.e., $\nabla_{x_y} M \equiv 0$ by design), the derivative with respect to $x_y$ is

$$\frac{\partial \Delta(x_y; \lambda)}{\partial x_y} = (1 - \lambda)\frac{\partial S_{\text{all}}}{\partial x_y} = (1 - \lambda)p_y. \tag{8}$$

This derivative confirms that $\lambda$ directly controls the monotonicity of the margin: $\lambda < 1$ (CE-like): the margin expands (positive gradient). $\lambda = 1$ (calibration point): the margin is decoupled from confidence (zero gradient), correcting the expansive property. $\lambda > 1$ (aggressive): the margin shrinks (negative gradient), enforcing a tighter boundary as confidence increases.

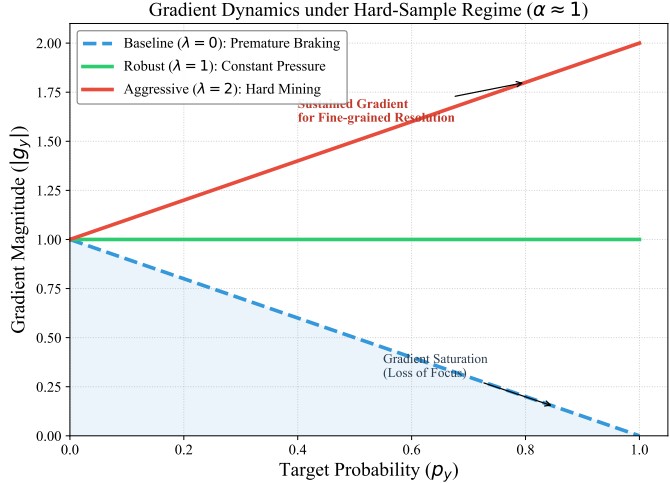

Figure 1: Gradient magnitude $|g_y|$ versus confidence $p_y$ for hard samples ($\alpha \approx 1$) under different $\lambda$, illustrating "delayed braking".

## 2.3 ANALYSIS OF GRADIENT DYNAMICS

Finally, we derive the exact gradient dynamics. Let $\alpha = \sigma\big(L_{\text{left}} - L_{\text{right}}\big)$ be the gating factor, where $\sigma(\cdot)$ is the sigmoid function. The total gradient $g_y = \frac{\partial \mathcal{L}_{\text{GDS}}}{\partial x_y}$ is given by

$$g_y = \alpha \frac{\partial L_{\text{left}}}{\partial x_y} + (1 - \alpha)\frac{\partial L_{\text{right}}}{\partial x_y}. \tag{9}$$

First, we explicitly compute the gradient of the hard constraint. Since $\frac{\partial S(M)}{\partial x_y} = 0$ (i.e., $\nabla_{x_y} M \equiv 0$ by design),

$$\frac{\partial L_{\text{left}}}{\partial x_y} = \lambda(0) + (1 - \lambda)p_y - 1 = (1 - \lambda)p_y - 1. \tag{10}$$

Recall that $\frac{\partial L_{\text{right}}}{\partial x_y} = p_y - 1$. Substituting these into the total gradient equation yields

$$g_y = \alpha[(1 - \lambda)p_y - 1] + (1 - \alpha)[p_y - 1]. \tag{11}$$

Simplifying this expression, we obtain the generalized gradient dynamics equation:

$$g_y = p_y(1 - \alpha\lambda) - 1. \tag{12}$$

We visualize the gradient magnitude $|g_y|$ as a function of prediction confidence $p_y$ for hard samples ($\alpha \approx 1$) in Fig. 1. **Blue (baseline, $\lambda = 0$):** the gradient decays linearly, causing premature saturation ("braking") even when the boundary is not yet sharp. **Green (robust, $\lambda = 1$):** the gradient remains constant and is decoupled from confidence. **Red (aggressive, $\lambda = 2$):** the gradient magnitude increases with confidence, effectively reducing the margin to force the model to mine harder features.

This equation reveals a tunable "delayed braking" mechanism.

## 3 EMPIRICAL ANALYSIS

In this section, we empirically validate the proposed framework using ViT-B/16 Dosovitskiy et al. (2021). We report a dataset-dependent divergence in optimal optimization regimes, revealing how margin dynamics interact with the specific topology of fine-grained manifolds.

### 3.1 EXPERIMENTAL SETUP

**Datasets.** We evaluate on three standard FGVC benchmarks:

Table 1: **Top-1 Accuracy (%) on FGVC Benchmarks.** The optimal $\lambda$ shifts based on data topology: **robustness** ($\lambda = 1$) favors noisy biological data (CUB), while **aggressiveness** ($\lambda = 2$) favors rigid geometric data (Cars).

| Dataset | CE (Baseline) | $\lambda = 0.5$ | $\lambda = 1.0$ | $\lambda = 1.5$ | $\lambda = 2.0$ |
|---|---|---|---|---|---|
| CUB-200 | 90.93 | 90.84 | **91.26** | 91.14 | 90.70 |
| Stanford Dogs | 89.71 | 89.43 | 89.92 | **90.07** | 89.97 |
| Stanford Cars | 90.30 | 90.22 | 90.47 | 90.72 | **90.92** |

- **CUB-200-2011** Wah et al. (2011).
- **Stanford Dogs** Khosla et al. (2011).
- **Stanford Cars** Krause et al. (2013).

**Implementation.** We utilize a ViT-B/16 backbone pre-trained on ImageNet-21k. We compare standard Cross-Entropy (CE) against our Generalized Dual-Scale Loss with $\lambda \in \{0.5, 1.0, 1.5, 2.0\}$. We sweep $\lambda \in \{0.5, 1.0, 1.5, 2.0\}$ (with $\tau = 1$) and keep all other hyperparameters fixed.

### 3.2 Results: The Rigidity-Aggressiveness Spectrum

Table 1 details the Top-1 accuracy across datasets. We observe that the optimal margin dynamic is not universal but shifts based on the nature of the data.

**1. Biological manifolds favor robustness (CUB-200).** On CUB, performance peaks at the **constant-margin regime ($\lambda = 1.0$)** (+0.33%) and degrades as optimization becomes more aggressive.

*Analysis.* Biological images often contain clutter (e.g., branches) and deformable features. While standard CE saturates too early, the aggressive hard mining of $\lambda = 2$ likely forces the model to overfit to background noise or ambiguous non-target features. $\lambda = 1$ strikes the optimal balance: it prevents gradient saturation (fixing the CE flaw) without introducing the instability of aggressive mining.

**2. Rigid manifolds favor aggression (Stanford Cars).** On Cars, we observe a monotonic improvement as $\lambda$ increases, with the **aggressive regime ($\lambda = 2.0$)** achieving the best result (+0.62%).

*Analysis.* Unlike birds, cars are rigid objects where differences are extremely subtle and geometric rather than textural. Here, geometric subtlety dominates over semantic hierarchy. The early braking of CE is detrimental because the model easily learns the coarse shape and stops. The delayed braking of $\lambda = 2$ appears necessary to force the optimizer to continue resolving minute structural discrepancies.

**3. The intermediate case (Stanford Dogs).** Dogs, sharing characteristics of both biological texture and significant pose variation, find their optimum at $\lambda = 1.5$ (+0.36%), bridging the gap between the two extremes.

### 3.3 Discussion

Across all datasets using the ViT backbone, the hybrid regime $\lambda = 0.5$ consistently underperforms standard CE, suggesting that partial correction is insufficient to overcome the intrinsic margin expansion. More importantly, our results indicate that there is no universal margin dynamics. For rigid, low-noise tasks, aggressive gradient maintenance ($\lambda > 1$) is beneficial; for noisy, high-variance tasks, a robust constant margin ($\lambda \approx 1$) is preferred.

## 4 Conclusion

We reinterpret CE through a geometric lens and identify a confidence-dependent intrinsic margin that can prematurely saturate gradients in fine-grained regimes. We propose the Generalized Dual-Scale (GDS) loss to continuously control margin dynamics via $\lambda$, bridging probabilistic smoothing and hard geometric constraints. Experiments with Vision Transformers indicate that optimal dynamics are topology-dependent: rigid manifolds benefit from more aggressive settings ($\lambda > 1$), while noisy manifolds favor robust constant margins ($\lambda \approx 1$), motivating topology-aware optimization.

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
