# OpenReview forum: "Generalized Dual-Scale Optimization: Topology-Aware Margin Dynamics in Fine-Grained Vision"
_ICLR.cc/2026/Workshop/Sci4DL — Sci4DL 2026_

### Official Review · Reviewer_DbJr · 2026-02-27

**Fit:** 2
**Significance:** 2
**Confidence:** 1

**Summary:**

As far as I understand, this paper addresses an issue of using cross entropy when learning labels from images of classes that are very similar. The learning seems to slow down prematurely and the paper introduces a new Generalized Dual-Scale (GDS) loss that addresses this issue. However, it is not one size fits all and there is a parameter $\lambda$ to fine tune the loss to the class of objects being classified.

**Strengths:**

New loss that addresses an issue with cross-entropy loss. The paper includes detailed experiments on how this loss improves training for a variety of classes and makes recommendations on how to chose $\lambda$.

**Suggestions:**

I have to admit I did understand very little of this paper. I tried to clarify things by using Gemini 3.1 Pro to explain concepts to me and search the literature for concepts I am not familiar with, but despite this effort, I do not have a clear idea what the contribution of the paper is. While I appreciate the need to have clear terminology, the introduction and description loses me in jargon. I am also not sure if terms like "intrinsic margin", "braking" and "buffer" are established concepts in the wider ML literature or whether they are particular to this paper. In either way, it would help me if they were clearly defined. Even if some of these definitions and details were moved to an appendix. Overall, I think it would be helpful if there was a high-level description of approach, etc. without resorting to jargon and then have a more in depth analysis description that uses formal definitions. All terms and definitions should be included in the paper, possibly in the appendix to stay within the page limit.

---

### Official Review · Reviewer_p9SW · 2026-02-27

**Fit:** 3
**Significance:** 2
**Confidence:** 2

**Summary:**

The paper studies the optimization dynamics of Cross-Entropy loss in fine-grained recognition. The authors argue that Cross-Entropy loss induces an implicitly expanding margin that leads to premature gradient saturation during training. To address this issue, they propose the Generalized Dual-Scale Loss, a unified objective that interpolates between standard Cross-Entropy loss and a hard-target constraint through a controllable parameter. Experiments on several FGVC benchmarks suggest empirical benefits of this approach, while indicating that the optimal balance between the two terms depends on dataset topology.

**Strengths:**

- The paper poses a clear scientific question regarding the optimization behavior of Cross-Entropy in fine-grained tasks and provides a intuitive theoretical arguments supporting their motivation.
- The proposed method is conceptually simple and can be easily adapted into existing training pipelines.

**Suggestions:**

- An empirical evaluation of whether the proposed approach reduces the intrinsic margins of the models compared to the baseline would further strengthen the paper.
- Additional qualitative analysis (e.g., visualization of failure cases) would improve the presentation and help better connect the optimization dynamics to the observed behavior.
- The abstract could be made more accessible by explaining the intuition behind the parameter $\lambda$ before presenting the formal notation.

---

### Official Review · Reviewer_YzND · 2026-02-28

**Fit:** 3
**Significance:** 2
**Confidence:** 2

**Summary:**

The paper introduces Generalized Dual-Scale (GDS) optimization, which controls confidence-dependent margin/gradient pressure by fusing standard cross-entropy with an adaptive hard constraint whose normalizer is a $\(\lambda\)$-weighted (convex) combination of the standard log-sum-exp energy $\(S_{\text{all}}\)$ and a hard-negative–anchored energy $\(S(M)\)$ built from $\(M=\max_{j\neq y}x_j\)$; the standard CE term and this adaptive hard constraint are then fused via a log-sum-exp (soft-maximum) gate.
Importantly, the maximum non-target logit $M$ is detached from the gradient graph, which means the parameter lambda effectively controls the "intrinsic margin" of the classifier during training.

The above statements are supported by theoretical analyses, and the proposed training procedure is empirically tested by applying ViT's to three standard (if small-scale) image classification problems.

**Strengths:**

- The core mechanism introduced is both clear, and the analysis thereof suggests to me this is an interesting idea to pursue.

**Suggestions:**

- The connection between the optimal lambda and the "topology" of the dataset seems thin to me. I think that crafting a parametrically controlled synthetic dataset (or, family of datasets) where this claim can be more explicitly tested would be a great addition to the next iteration of this work.
- Empirical evaluations are somewhat lacking (note, I think if the above suggestion is executed well, you might not even need these experiments!):
  - The effect sizes in Table 1 seem very small. It would be good to get a sense for what run-to-run variability is (over different random initializations) for these tasks and datasets in order to contextualize this.
  - There should probably be comparisons to related methods (some of which are cited) in addition to standard cross-entropy to facilitate an evaluation of the impact and novelty of this work.

---

### Meta-Review · Area_Chair_LoLV · 2026-03-01

**Recommendation:** Accept

**Metareview:**

The paper is well written, easy to follow and makes significant contributions. I recommend acceptance.

---

### Decision · Program_Chairs · 2026-03-02

Accept